# Physical activity and sedentarism among seniors in France, and their impact on health

**Jérémy Pierre** [1]*, **Cécile Collinet**[1], **Pierre-Olaf Schut**[1], **Charlotte Verdot** [2]

**1** Sport Sciences, Gustave Eiffel University, Paris, France, **2** Nutritional Surveillance and Epidemiology Team (ESEN), Direction of Non-Communicable Diseases and Trauma (DMNTT), Santé Publique France (French Public Health Agency), Sorbonne Paris Nord University, Epidemiology and Statistics Research Center–University of Paris (CRESS), Bobigny, France

* jeremy.pierre@univ-eiffel.fr

## Abstract

### Background

In the context of the ageing of the French population, physical activity becomes a principal means for maintaining good health. International organisations are thus giving increasing importance to physical activity in programmes of disease-prevention. In parallel with these concerns, studies have shown the impact of sedentary activities (in particularly as a result of the seated position and screen time) on health.

### Objective

To show the links between physical activity, sedentarism and health indicators and to identify the socio-demographic variables by which they are influenced (particularly gender).

### Study design

This is a transversal epidemiological study conducted among the French population between 2014 and 2016 by Santé publique France, the national public health agency.

### Methods

The RPAQ (Recent Physical Activity Questionnaire) was used to measure the physical activity and sedentary lifestyle of individuals. The analyses focus on the behaviours among the population of older adults (55–74 years old, n = 1155).

### Results

A third of French older adults does not meet the WHO recommended physical activity levels, particularly so among women. The results of this survey point to significant links between health indicators (especially overweight and obesity) and the physical activity level and sedentarity of older adults. From the age of 60, energy expenditure linked to physical activity increases before decreasing from the age of 65. Older adults spend almost 6 hours per day in sedentary activities. The combinations between physical activity and sedentarism

**Data Availability Statement:** Data Availability Statement: The data is not deposited in publicly available repositories but are available on request from Santé publique France. For more information, please consult: https://www.santepubliquefrance.

fr/les-actualites/2019/appel-a-projet-d-ouverture-des-bases-de-donnees-y-compris-de-la-collection-biologique-de-l-etude-esteban-2014-2016. Accessed on: 22 May 2022.

**Funding:** The authors received no specific funding for this work.

**Competing interests:** The authors have declared that no competing interests exist.

highlight four profiles of older adults. The most active profile is the one with the best health indicators.

## Conclusion

The links between health, sedentarity and physical activity are unequivocal: people who achieve the WHO recommendations for physical activity and spend less than 7 hours each day in sedentary activities are those who also have the best health indicators. These results vary with sociographic characteristics and reveal significant links with health indicators.

## 1. Introduction

Physical activity (PA) is a growing element in public policies insofar as scientific research has demonstrated its protective role in the prevention of non-communicable diseases. "*Physical activity is defined as any bodily movement produced by skeletal muscles that require energy expenditure*" [1]. It thus covers many forms of activity, ranging from sport to forms of personal travel and domestic and leisure activities. Any attempt to measure an individual's overall activity must take into consideration all of these components. This is all the more the case if we take into account public health recommendations expressed in terms of an individual's total volume of activity. Thus, for adults, as for older adults, it is recommended to "*do at least 150 minutes of moderate-intensity aerobic physical activity throughout the week or at least 75 minutes of vigorous-intensity aerobic physical activity throughout the week, or an equivalent combination of moderate- and vigorous-intensity activity*" [2]. The literature as a whole bears out the positive effects of physical activities on the different health dimensions [3]. Those individuals who do not meet these recommendations are considered physically inactive, which is a risk factor in the development of non-communicable diseases, such as coronary heart disease, type 2 diabetes or certain cancers [4–10].

Sedentary behaviour (SB) is defined as "*any waking behaviour characterised by an energy expenditure of less than or equal to 1.5 METs (Metabolic Equivalent Task) while in a sitting or reclining posture*"[11]. It is thus possible to be physically active by means of regular physical activity while being sedentary through long periods of very low energy expenditure, particularly due to prolonged sitting. A sedentary lifestyle is also– independently of PA– a risk factor in non-communicable diseases [12–18]. In order to enjoy greater health benefits, it is therefore necessary for individuals to be both physically active and to limit their sedentary behaviours. These recommendations remain valid throughout life and are even more important with advancing age, which is generally associated with increasing vulnerability [19].

Getting the public to be active, older adults in particular, is an important issue in the social and cultural context of our ageing societies [20]. Many studies on this theme have appeared in recent years in the form of books [21–23] and special issues of journals (*Ageing & Society* 2012; *Sport, Athleticism, Activity* 2012; *Gerontology and Society* 2018). These studies point to the positive effect on health of being physically active in old age [24] in tandem with the limitation of sedentary behaviours. Moreover, analysis of texts from leading international organisations [25] reveals that, for the past twenty years, PA has been considered a central plank in the prevention of the detrimental effects of ageing alongside the growing importance of the conception of active ageing as a model [26, 27]. This subject is of interest to most governments and affects all aspects of collective life.

Many studies show that PA can play an important role in the prevention and mitigation of many of the deteriorations associated with biological ageing [28]. PA has become one of the key arms in the fight against ageing and is a pillar in governmental strategies to improve health throughout life [2, 29]. International surveys reveal a continuous decline in the level of activity with advancing age, in particular once retirement age is reached, of which the average age is 62.7 years in France. Statistics demonstrate that the levels of PA among the older adults are very low, which may suggest that individuals would be less likely to take up a PA or remain physically active as they age for a variety of reasons related to individual behaviours and social determinants [30]. Thus, PA levels among older adults are insufficient to ensure good health is maintained [31, 32]. Studies exist in social psychology that encourage French older adults to practise PA as part of specific programmes [33], and others that measure the effects of programmes on different health indicators [34, 35], but no study exists on the general measurement of PA and sedentarism in this particular age group. This article has the objective of analysing activity levels among French senior citizens in terms of both physical activity (domestic, leisure and sports) and sedentary behaviours (seated professional activity and screen time). Rates of physical activity and sedentarism will be cross-referenced to highlight four profiles of seniors.

## 2. Materials and methods

### 2.1 Data source

The data in this article are taken from the "*health study on environment, bio-monitoring, physical activity and nutrition*" (Esteban) carried out between April 2014 and March 2016 by Santé Publique France, the national public health agency.

**2.1.1 Institutional review board statement.** The Esteban study was approved by the Consultation Committee for the Protection of Participants in Biomedical Research of "Ile-de-France IX" (no. 2012-A00459-34); the computer processing of these data obtained authorization from the Council of State (Council of State decree n˚2013–742 published in the official journal on 14 August 2013) after approval of the French National Information and Citizen Freedom.

**2.1.2 Informed consent statement.** Written informed consent was obtained from all subjects involved in the study.

Esteban is a cross-sectional epidemiological study that is representative of the French population as a whole. Its objectives were to estimate the levels of exposure to environmental substances, and to monitor chronic diseases and nutrition-related matters (food consumption, nutritional status, physical activity and sedentary lifestyle) of the French. The study protocol included a questionnaire (self-administered in the presence of a researcher), a dietary survey and a health examination. These studies received the approval of the Advisory Committee on Information Treatment in the field of Health Research (CCTIRS), the French Data Protection Authority (Cnil) and the Personal Protection Committee (CPP). All participants signed informed consents.

The sample, given by a three-stage stratified random sample design, was composed of 2678 adults aged between 18 and 74, representative of the French population. This article focuses on the analysis of people aged 55 to 74 (n = 1155) for whom sedentary behaviour and physical activity can be central factors with regard to poor health. The total population numbers just over 15 million people in France.

### 2.2 Measuring PA and sedentary behaviours (SB)

The RPAQ (*Recent Physical Activity Questionnaire*) [36, 37] was used. This questionnaire enables assessment of the daily physical activity and sedentary behaviours of adults during the previous four weeks. It includes questions on physical leisure and sports activities (frequency

**Table 1. Levels of physical activity from the WHO.**

| Low | Absence of PA or level of PA that does not allow recommendations to be achieved |
|---|---|
| Moderate | 3 days or more per week of vigorous-intensity PA of at least 25 minutes/day |
| | OR 5 days or more per week of moderate-intensity PA of at least 30 minutes/day |
| | OR 5 days or more per week of moderate- or vigorous-intensity PA that allows a minimum of 600 METs minutes/week to be achieved |
| Intense | 3 days or more per week of vigorous-intensity PA that allows a minimum of 1500 METs minutes/week to be achieved |
| | OR moderate- or vigorous-intensity PA each day of the week that allows a minimum of 3000 METs minutes/week to be achieved |

and duration), and activities performed in the home (television, computer, climbing stairs, etc.) and at work (quantity and type of work, home-work journeys, etc.). In the Esteban study, additional questions were asked on household activities (housework, gardening, DIY, etc.). Data analysis was carried out taking into account the duration and frequency of each activity, and its intensity expressed in terms of its metabolic equivalent (Metabolic Equivalent Task–MET). An energy expenditure score has been determined for each activity [38]. Several indicators were created: the energy expenditure score in relation to physical activities (expressed in METs minutes/week), the duration of sedentary activities (expressed in hours per day), the overall level of physical activity in relation to the achievement or not of the WHO recommendations (Table 1) and the overall level of sedentarism (Table 2).

### 2.3 Socio-demographic data and health

The survey also includes a socio-demographic dataset collected during a face-to-face interview. These relate to the family situation (whether living in a couple or not, with or without children), educational level (lower than, higher than or equal to the French high-school diploma), profession and socio-professional category (SPC) and whether the individual performs a professional or voluntary activity.

Several types of health data were also collected as part of a health examination, in particular a measure of body mass index (BMI), the reporting of a long-term illness (ALD: it is a major or long-term illness for whose health costs the State accepts responsibility), the perceived state of health, the presence of chronic pathologies such as diabetes, hypercholesterolemia or cardiovascular diseases, and the consumption of tobacco and alcohol.

### 2.4 Statistical analysis

The set of analyses was performed on data weighted and adjusted using the Stata 14® software. The complex sampling design [39] as factored in particularly when estimating the variances and 95% confidence intervals [CI 95%] using Stata 14's "svyset" function. The Wald and Pearson's chi-squared tests were used to determine the existence of a significant association between two variables. The following significance thresholds were used: $^*p<0.05$; $^{**}p<0.01$; $^{***}p<0.001$.

**Table 2. Levels of sedentarism (based on the number of hours spent daily on sedentary activities ($<1.6$ METs)) from the French national observatory for physical activity and Sedentary Behaviours (SB).**

| Low sedentarism | Duration of the sedentary activities ($< 1.6$ METs) $< 3$ hours/day |
|---|---|
| Moderate sedentarism | Duration of the sedentary activities ($< 1.6$ METs) 3–7 hours/day |
| High sedentarism | Duration of the sedentary activities ($< 1.6$ METs) $> 7$ hours/day |

## 3. Results and discussion

### 3.1 Presentation of the study population

Table 3 gives the characteristics of the study population composed of 1155 adults aged between 55 and 74. This sample is made up of 51.7% females and 48.3% males with a mean age of 63.0 years. Almost all the older adults lived without their children (96.6%) and the great majority lived as part of a couple (80.4% of the men and 68.3% of the women). One third had an educational diploma equal to or higher than a high-school leaver's diploma (*baccalauréat*) and the socio-professional category (SPC) that was most represented was that of employees, intermediate occupations and workers, which is representative of the data for the French population aged 50 or more (INSEE, continuous survey employment, 2019). Almost 4 older adults out of 10 regularly performed a professional and/or voluntary activity.

**Table 3. Socio-demographic and health data of the study population (55–74 years)–Esteban study 2014–2016.**

| | Total n = 1155 | Men n = 509 | Women n = 646 | p* |
|---|---|---|---|---|
| Breakdown (%) | | 48.3 | 51.7 | |
| Age (mean, sd) | 63.0 (0.2) | 62.7 (0.3) | 63.3 (0.3) | ns |
| Living in a couple (%) | 74.1 | 80.4 | 68.3 | <0.001 |
| Living with children (%) | 3.4 | 5.5 | 1.4 | <0.001 |
| Educational diploma ≥ high-school (%) | 33.0 | 35.2 | 30.9 | ns |
| SPC (%) | | | | |
| Farmers | 2.4 | 3.3 | 1.6 | <0.001 |
| Craftsmen, tradesmen | 6.3 | 9.1 | 3.7 | |
| Managers and higher grade prof. | 10.5 | 15.7 | 5.7 | |
| Intermediate occupations | 27.0 | 31.6 | 22.7 | |
| Employees | 32.3 | 14.8 | 48.6 | |
| Workers | 19.2 | 24.7 | 13.9 | |
| Don't know | 2.3 | 0.8 | 3.8 | |
| Professional or voluntary activity (%) | 39.5 | 41.7 | 37.4 | ns |
| BMI (%) | | | | |
| Underweight <18.5 | 1.9 | 2.0 | 1.8 | <0.001 |
| Normal [18.5–25.0] | 39.5 | 30.5 | 47.9 | |
| Overweight [25.0–30.0] | 37.0 | 45.8 | 28.7 | |
| Obese ≥ 30.0 | 21.6 | 21.7 | 21.6 | |
| ALD (%) | 26.9 | 31.8 | 22.4 | <0.01 |
| Chronic health problem (%) | 49.5 | 49.2 | 49.8 | ns |
| Functional limitations (%) | | | | |
| Strongly limited | 7.2 | 7.4 | 7.1 | ns |
| Limited, but not strongly | 20.6 | 17.1 | 24.0 | |
| Perceived health (%) | | | | |
| Very good | 20.9 | 23.8 | 18.2 | 0.05 |
| Good | 50.4 | 49.0 | 51.6 | |
| Reasonable | 24.0 | 24.7 | 23.5 | |
| Bad | 4.2 | 2.2 | 6.1 | |
| Very bad | 0.4 | 0.3 | 0.5 | |
| Don't know | 0.1 | 0 | 0.1 | |
| Daily consump. of tobacco (%) | 13.1 | 18.1 | 8.5 | <0.001 |
| High alcohol consumption (%) | 6.9 | 13.1 | 1.2 | <0.001 |

* p = value of the difference between men and women

With regard to the health data, the majority of the sample were overweight, among whom 21.6% were obese. Three men and two women out of 10 stated that they were being treated for a long-term illness and nearly half reported that they suffered from a chronic problem (49.5%). One in four people said they were faced by functional limitations in their daily life, 7.2% of which were major, a situation that can lead to becoming overweight and unhealthy eating behaviours [40]. However, few reported feeling in poor health (only 4.6%), with the great majority saying they were in good or very good health (71.3%). Lastly, more men than women reported risky behaviours as 18.1% stated that they smoked tobacco everyday (against 8.5% of women) and 13.1% reported high alcohol consumption (compared with 1.2% of women).

## 3.2 Physical activity levels vary with gender

The results show that more than one in three older adults do not meet the WHO recommendations for physical activity (Table 4): those concerned are 28.0% of men and 42.5% of women between the ages of 55 and 74. These figures are comparable internationally. Currently, the incidence of physical inactivity in the adult population is 36.8% in Western countries around the world (31.2% for men and 42.3% for women [41]), and particularly so for the seniors [30–32, 42]. The level of vigorous-intensity physical activity (this corresponds to 3 or more days of vigorous PA per week, making it possible to achieve a minimum of 1500 METs minutes/week; or a moderate or vigorous PA each day of the week, giving a minimum of 3000 METs minutes/week) is also low for this age category (11.0% for men and 3.7% for women).

The mean energy expenditure for domestic PA was 2495.5 METs minutes/week for men and 1760.0 METs minutes/week for women. The expenditure concerning active transport (these data only concern those people who stated they used active transport (walking, cycling) to go to work: older adults who do not use these methods are not taken into consideration here) was 106.4 METs minutes/week for the men and 47.7 METs minutes/week for the women who use these methods of active transport. Lastly, the mean energy expenditure for sports and leisure PA was 2201.4 METs minutes/week for men and 1469.8 METs minutes/week for women (Table 5).

Among women, this score did not alter substantially with age (it increased up to the age of 63). In contrast, there is a clear increase among men from the age of 61 (Fig 1). This increase in practice among men in their 60s has been observed in other studies in France [43]; it might be supposed that the lack of change in women's activity levels is due to the fact that they stop work earlier than men and that there is some form of continuity in their lifestyle.

The increase among men between 55 and 63 years may be related to their leaving the world of work and to the consequent increase in their free time that they use for recreation and PA, as has been shown to occur in studies of lifestyle among older adults once they retire [44].

**Table 4. Physical activity levels for those aged 55–74.**

|  | Total | Men | Women | p* |
|---|---|---|---|---|
| Those meeting WHO recomm. levels (%) | 64.5 | 72.0 | 57.5 | <0.001 |
| Level of physical activity (%) |  |  |  |  |
| Low (below recommended levels) | 35.5 | 28.0 | 42.5 | <0.001 |
| Moderate | 57.3 | 61.0 | 53.8 |  |
| Intense | 7.2 | 11.0 | 3.7 |  |
| Energy expenditure due to PA in METs minutes/week | 3136.7 | 4314.5 | 2037.0 | <0.001 |

* p value of the difference between men and women

**Table 5. Energy expenditure for men and women for different types of PA.**

| Type of physical activity | Activities | Mean METs minutes/week | | p* |
|---|---|---|---|---|
| | | Men | Women | |
| Domestic physical activities | Housework | 418.1 | 1228.5 | <0.001 |
| | Gardening | 981.6 | 400.4 | <0.001 |
| | DIY | 1129.0 | 166.9 | <0.001 |
| | Total | 2527.9 | 1795.7 | <0.001 |
| Methods of active transport | Walking | 39.6 | 39.1 | ns |
| | Cycling | 66.8 | 8.6 | ns |
| | Total | 106.4 | 47.7 | ns |
| Sports and leisure physical activities (only the most commonly practised activities are included in the table) | Walking | 862.3 | 864.9 | ns |
| | Cycling | 404.4 | 89.7 | <0.001 |
| | Gymnastics, fitness | 180.0 | 209.1 | ns |
| | Water sports | 97.7 | 177.0 | ns |
| | Mountain sports | 170.7 | 17.4 | <0.01 |
| | Hunting and fishing | 136.8 | 1.1 | <0.01 |
| | Running | 75.2 | 15.0 | <0.001 |
| | Dancing | 28.0 | 49.8 | ns |
| | Golf | 54.4 | 12.0 | ns |
| | Racquet sports | 36.0 | 8.1 | ns |
| | Martial arts | 37.3 | 4.8 | ns |
| | Bowls and bowling | 35.2 | 1.6 | <0.01 |
| | Body building | 19.1 | 5.2 | <0.05 |
| | Team sports | 20.4 | 0.18 | ns |
| | Total leisure Pas | 2201.4 | 1469.8 | <0.001 |

However, in the Esteban study, no difference is seen in the level of PA among older adults whether or not they pursue a professional/voluntary activity.

As they approach the age of 70, however, a deterioration in their state of health [45] might explain the decrease in their practice of PAs. Several studies have shown that the primary reason given by older adults for the non-practice of PAs is linked to health problems [43].

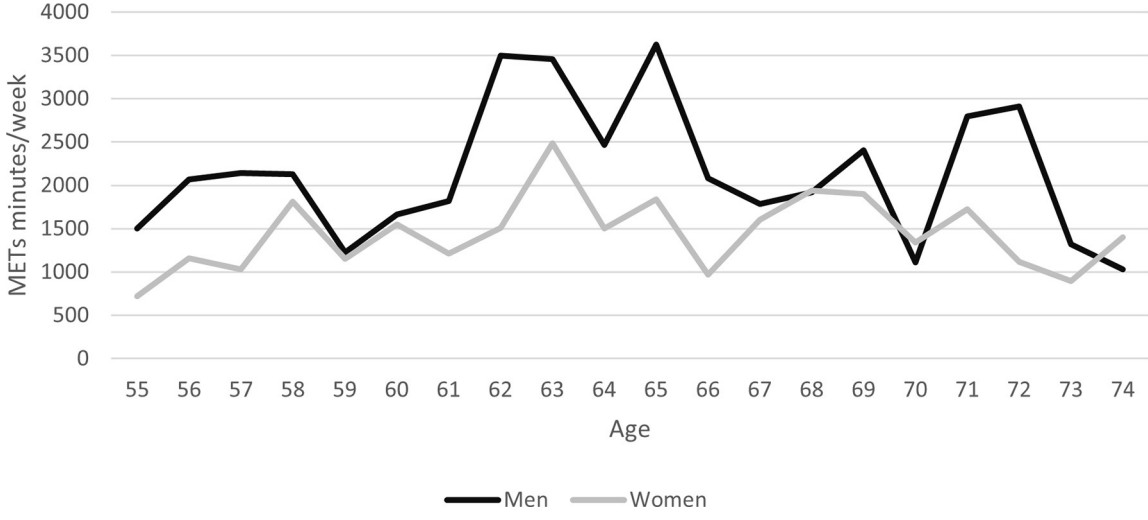

**Fig 1. Energy expenditure in METs minutes/week based on the cumulation of PA (sports and leisure physical activities, active transport, domestic physical activities) by gender.**

In addition to having an impact on the overall levels of PA, gender is an important discriminating factor in the activities performed. Regarding domestic activities, men put significantly more effort into gardening and DIY activities, whereas women are significantly more involved in household activities (Table 5). In sports and physical activities (SPAs), the results evince trends already seen in the national survey on the physical and sports practice of the French [46], such as the group of SPAs most practised by the French– walking, cycling and water sports– with this latter category preceded by gymnastics and fitness activities. Men are more involved in cycling, mountain sports, hunting, fishing and running, while women more specifically practice gymnastics, fitness activities and water sports.

### 3.3 A PA level linked to methods of transport, job type and state of health

The results show that there is a significant link between the most commonly used methods of transport (the data used here refer to active transport used for other than work-home journeys) and the level of PA (p<0.01). More specifically, individuals who meet the WHO recommended PA levels are more likely to walk or cycle to work than others, particularly among women (p>0.05). Reciprocally, the use of a motor vehicle is more frequent among those people with a low PA level.

Social milieu is also a strong indicator of PA practice. Considered with regard to the level of education and SPC, this observation is confirmed here. The higher the level of education, the higher the levels of moderate- and vigorous-intensity PA (p<0.05). The SPCs of managers and intermediate occupations have the highest average scores for energy expenditure related to sports and leisure PA, while the lowest scores are those of craftsmen and farmers. These data provide confirmation of both French and international studies on the subject.

The results of this survey also point to significant links between health indicators and the PA level of older adults. First, individuals who report they are in very good health are more likely to practise vigorous-intensity PA; conversely, those who consider themselves to be in poor health mostly have a low level of PA (p<0.001). The greater the deterioration in perceived health, the greater the increase in low PA level. This observation is also found in the study on the impact of PA on the health of older adults [47]. Note that research has shown that it is more the nature of the activities that influences perceived health than the quantity of energy actually expended [48].

PA recommendations are most frequently met in the "normal" category of BMI (18.5–24.9). It is in the "overweight" and, even more so, the "obese" categories that the percentage of those meeting PA recommendations is lowest, particularly so among women. Moreover, people with ALD have the lowest levels of PA (p<0.05). For those who suffer from chronic problems, 57.6% of those individuals with low PA are affected compared with 38.3% of those with high PA (p<0.05). If these figures to some extent confirm the protective role played by PA on health [49], they may also suggest that a person suffering from ALD or a chronic problem is less able to perform vigorous-intensity PA. In this sense, LaMonte et al. [50] show that even a low PA level contributes to good health, something that the WHO also recommends.

## 4. Almost 6 hours per day spent in sedentary activities

### 4.1 Screens: An activity popular among older adults

The battle against a sedentary lifestyle has become a major concern for public health authorities [51]. In addition to the lack of physical activity, sedentarism is a risk factor for non-communicable diseases [12, 14, 18, 52]. The PA of the seniors is not exempt from this trend, which affects all sections of the French population [53]. In the Esteban survey, 88.9% of the older adults evaluated had a moderate or high degree of sedentary lifestyle and 28.3% spent more

**Table 6. Time spent each day on sedentary activities by gender.**

| Activities | Time spent each day on sedentary activities | | | p* |
|---|---|---|---|---|
| | Average | Men | Women | |
| Television | 3h39 | 3h32 | 3h46 | ns |
| Computer (non-professional activity) | 1h18 | 1h24 | 1h12 | ns |
| Video games | 0h07 | 0h03 | 0h11 | ns |
| Other non-screen sedentary activities | 0h46 | 0h51 | 0h41 | ns |

than 7 hours per day in sedentary activities irrespective of gender. More specifically, older adults spent an average of 5:50 hours per day in sedentary activities, including 5:04 hours in front of a screen (Table 6). The time spent in front of a screen includes the time before a television, games console and computer (though not as part of a professional activity).

## 4.2 The influence of professional/voluntary activity on the degree of sedentarism

People who exercise a professional/voluntary activity spend 6:18 hours per day in sedentary activities, compared to 5:32 for those who do not. A hasty interpretation of this might lead to the conclusion that doing a professional/voluntary activity generates an increase in sedentarism. This is true when the professional/voluntary activity is sedentary and performed seated, which is the commonest situation (Table 7). However, while maintaining such an activity can help increase a sedentary lifestyle in some people, we should not forget the benefits associated with maintaining an activity with advancing age. Indeed, numerous studies have shown the health benefits of carrying out a regular professional or voluntary activity. Many ways exist to encourage older adults to become volunteers [54] and the resulting effects benefit both all of society [55] and the individual, through the maintenance of social contact in particular [56]. It is therefore necessary to find a right balance between these various activities through limiting the overall daily time spent in sedentary activities and acting more specifically on free or leisure time.

The degree of sedentarism is also linked to SPC. For people who are still working, it is those in the categories of managers, intermediate occupations and employees who spend the most time in sedentary activities, a long way ahead of farmers (Table 8). This distinction is a reflection of their working conditions: managers spend most of their working time seated [57] and therefore have a very high level of sedentarism.

Note that ceasing a professional activity brings significant change for these SPCs: those in the grouping mentioned above reduce the time they spend on sedentary activities (especially managers) while farmers increase that time by almost 2 hours per day.

**Table 7. Time spent each day on sedentary activities broken down by type of work (for people in a professional/ voluntary activity, n = 477, i.e. 39.5% of the respondents).**

| | Type of work | Nos. | Time spent each day on sedentary activities |
|---|---|---|---|
| People with a professional/voluntary activity | Sedentary | 267 | 8h07 |
| | Standing | 141 | 4h29 |
| | Manual activity | 61 | 4h47 |
| | Intense manual activity | 8 | 4h48 |
| | Average | 477 | 6h18 |
| People with no professional activity | | 651 | 5h32 |
| Average for the whole population | | 1128 | 5h50 |

Table 8. Time spent each day on sedentary activities by SPC.

| SPC | % of the total population | Time spent each day on sedentary activities | |
|---|---|---|---|
| | | People still working | People no longer working |
| Farmers | 2.4 | 3h07 | 5h02 |
| Craftsmen, tradesmen and business leaders | 6.3 | 5h28 | 5h49 |
| Managers and higher-grade professionals | 10.5 | 7h12 | 5h46 |
| Intermediate occupations | 27.0 | 6h35 | 5h22 |
| Employees | 32.3 | 6h19 | 5h24 |
| Workers | 19.2 | 5h21 | 5h23 |
| Don't know | 2.3 | 5h59 | 6h43 |
| Total | 100 | 6h18 | 5h32 |

Whereas an increase in sedentarism among young people is of great concern, in particular due to the screen time [58], the problem takes another form among older adults. As mentioned above, nearly one in three older adults experiences a high degree of sedentarism. Although this does not appear to impact perceived health, the results reveal that there is a higher prevalence of obesity (BMI>30) among those people who experience a high level of sedentarism (36.1% vs. 6.5% among those people with a low level of sedentarism).

# 5. Profiles

## 5.1 Differences in behaviours between men and women

Men's and women's behaviours differ significantly, as has been shown. Women practise fewer risky behaviours, but they are less active than men. Men are more active in sports and leisure activities [59] and use more active methods of transport. At home, women focus more on domestic activities, which require less expenditure of energy than the gardening or DIY activities preferred by men. Thus, men have a higher energy expenditure and attain health recommendations more often than women, as shown by Moschny et al. [60].

## 5.2 The 4 profiles, different combinations of PA and sedentary behaviours (SB)

The results have been presented on PA, then on the sedentary lifestyle of older adults. These analyses have made it possible to detail the behaviours of older adults while also revealing the influence of certain sociographic variables on activity levels and, even more so, on the nature of the activities performed. Given the influence of physical inactivity and sedentarism in the development of non-communicable diseases, and of the independence of these two risk factors, it is also interesting to study the distribution of these behaviours within specific profiles. There is no doubt that vigorous-intensity PA does not compensate for a high level of sedentarism and that the health risks are cumulative. The study by Chau et al. [12] also shows that every hour spent sitting over 7 hours per day increases the mortality rate by 5%, all causes combined (regardless of the compensation effect of physical activity).

The profiles are defined by the different combinations of the two dimensions: the degree of physical activity (the fulfilment or not of the recommended values) and the degree of sedentarism (more or less than 7 hours of sedentary activities per day). The clearly predominant profile in the 55–74 age group represents those people who are physically active and non-sedentary (50.3%, Table 9).

The three tables below allow a comparison to be made of the behaviours and their effects on health of the four PA & SB profiles. More specifically, the categories have been compared

**Table 9. PA & SB profiles of the 55–74 age group.**

| | PA & SB profiles | Description | % of total (95% CI) | Extrapolation (95% CI) |
|---|---|---|---|---|
| Profile 1 | Inactive and sedentary | PA level less than recommendations and duration of sedentary activities > 7h per day | 14.1 [11.5–17.2] | 2,232,887 [1,821,149–2,723,805] |
| Profile 2 | Inactive and not sedentary | PA level less than recommendations and duration of sedentary activities < = 7h per day | 21.4 [18.4–24.7] | 3,388,921 [2,913,839–3,911,512] |
| Profile 3 | Active and sedentary | PA level meets recommendations and duration of sedentary activity > 7h per day | 14.2 [11.8–17] | 2,248,723 [1,868,657–2,692,134] |
| Profile 4 | Active and not sedentary | PA level meets recommendations and duration of sedentary activity < = 7h per day | 50.3 [46.5–54.2] | 7,965,814 [7,363,777–8,583,155] |
| | Total | | 100% | 15,836,080 |

with domestic activities, sports and leisure activities (Table 10), sedentary activities (Table 11) and health data (Table 12).

## 5.3 The physically inactive. . .

This section concerns those people who are physically inactive, that is to say those who do not meet the minimum WHO recommendations in terms of physical activity. Unsurprisingly, they report very low volumes of physical activity, particularly in the "walking– swimming– cycling" set of activities (Table 10).

**5.3.1. . .and sedentary (14.1% of older adults).** A very significant link exists between this category and the fact of having a professional/voluntary activity (46.2% vs. 39.5% on average, p<0.01). The activity is predominantly performed in a seated position. The individuals in question prefer to move around using motor vehicles. Much of their sedentary lifestyle is due to the time they spend seated watching television, a very strong characteristic of this profile, the effects of which are harmful to health [61]. These individuals perform few domestic or sporting activities, except for a little walking. This profile reports the worst health data (Table 12), with prevalence of chronic problems, functional limitations and being overweight that are rising sharply. There is also a high rate of currently smokers (16.7%).

**5.3.2 . . .and non-sedentary (21.4% of older adults).** This category is the most feminized (67.3% vs. 51.7% on average, p<0.001). The frequency of individuals having a professional/voluntary activity is close to the average, but this profile specifically includes people who exercise their activity standing (90%, p<0.001), which is a factor in the decrease in their level of sedentarism. Aside from their physical inactivity, these people are not sedentary because they spend

**Table 10. Energy expenditure linked to PA by PA & SB profiles.**

| | Mean METs minutes/week | | | |
|---|---|---|---|---|
| Types of activities | Inactive and sedentary (P1) | Inactive and not sedentary (P2) | Active and sedentary (P3) | Active and not sedentary (P4) |
| Domestic physical activities | 1160.5 | 1189.1 | 2510.6 | 2732.7 |
| Sporting and leisure physical activities | 746.5 | 781.7 | 2190.1 | 2464.0 |

**Table 11. Daily time spent on sedentary activities by PA & SB profiles.**

| | Inactive and sedentary (P1) | Inactive and not sedentary (P2) | Active and sedentary (P3) | Active and not sedentary (P4) |
|---|---|---|---|---|
| Time spent each day on sedentary activities | 9h19 | 4h37 | 9h06 | 4h28 |

**Table 12. Health data by PA & SB profiles.**

| | All | Inactive and sedentary (P1) | Inactive and not sedentary (P2) | Active and sedentary (P3) | Active and not sedentary (P4) | p* |
|---|---|---|---|---|---|---|
| *Perceived health (%)* * | | | | | | <0.05 |
| Good | 95.4 | 92.6 | 91.7 | 97.9 | 97.0 | |
| Poor | 4.6 | 7.4 | 8.3 | 2.1 | 3.0 | |
| *ALD (%)* | | | | | | ns |
| Yes | 26.9 | 31.0 | 32.8 | 19.3 | 25.4 | |
| *Chronic problem* | | | | | | ns |
| Yes | 49.5 | 60.1 | 55.9 | 46.0 | 44.8 | |
| *Overweight and obese* | | | | | | ns |
| Yes (BMI > = 25) | 58.6 | 64.1 | 62.7 | 59.7 | 55.0 | |
| *Functional limitations* *** | | | | | | <0.001 |
| Yes | 27.9 | 45.0 | 33.9 | 17.3 | 23.5 | |
| *Tobacco* | | | | | | ns |
| Daily smoker | 13.1 | 16.7 | 12.3 | 16.9 | 11.5 | |
| Former smoker | 32.8 | 23.5 | 32.6 | 41.0 | 33.2 | |
| Non-smoker | 54.1 | 59.8 | 55.1 | 42.1 | 55.3 | |
| *Alcohol* | | | | | | ns |
| Consumption > the marker value (more than 20 grammes per day for women and 30 grammes for men) | 7.0 | 8.8 | 6.1 | 3.7 | 7.9 | |
| Consumption < = marker value | 85.6 | 83.0 | 81.4 | 92.2 | 86.2 | |
| Non-drinker | 7.4 | 8.2 | 12.5 | 4.1 | 5.9 | |

very little time in front of screens and are very active in certain domestic activities, such as washing-up and doing the laundry. These people are those whose perceived health is the worst and whose objective health indicators are not good, in particular regarding ALD (32.8%).

## 5.4 The physically active

The physically active are older adults who practise moderate or vigorous-intensity physical activity, in accordance with health recommendations, and who form the majority in the sample. These people thus have higher than average PA scores: 4700 METs minutes/week for the "physically active and sedentary" profile and 5197 METs minutes/week for the "physically active and non-sedentary" profile.

**5.4.1 . . .and sedentary (14.2% of older adults).** This profile has the highest proportion of older adults who practise a professional activity. They achieve the WHO recommendations for PA by performing domestic activities such as DIY, gardening and cutting the grass, and by walking and rambling. Nevertheless, in spite of having a satisfactory PA level, the time they spend each day in sedentary activities is very high (9:06 hours on average, Table 11), which is detrimental to maintaining good health. The data demonstrate that this group have good perceived health although the majority of them are overweight (59.7%). They suffer less from functional limitations (as ALD), which enables them to continue to practise physical activities in the home and in sports. In consequence, they maintain an adequate general state of health.

**5.4.2 . . .and non-sedentary (50.3% of older adults).** This profile, the most numerous among older adults, is also the most masculinized (55.7% vs. 48.3% on average, p<0.001). This category has the highest score of people without a professional activity (65.5% vs. 60.5% on average, p<0.01), suggesting that their extra free time is devoted to active leisure pursuits. More specifically, they are people who garden a lot and practise many physical and sports

activities, including walking, rambling, swimming and even floor exercises. In addition to being very active physically, this category spends the least time in sedentary activities and in front of screens. The effects on their health are unequivocal: while this profile perceives its health to be very good, the objective data suggest the same.

## 6. Conclusion

The results of this study show that half of the population of seniors is both active and non-sedentary (profile 4), an encouraging figure. Only a third of French older adults are physically inactive (profiles 1 and 2, Table 9), echoing a meta-analysis carried out in 122 countries that shows that 31.1% of adults in the world are inactive and that 41.5% of adults spend 4 or more hours seated each day [62].

Two results are particularly striking: first, the differences between men and women, and second, the impact of physical activity and sedentarism on the BMI.

The effects on health are unequivocal: people who achieve the WHO recommendations for physical activity and spend less than 7 hours each day in sedentary activities have the best health indicators (profile 4).

The pandemic has had a major impact on seniors' behaviours: physical activity has diminished and sedentarism increased [63]. It would be of interest to measure whether the levels of activity and sedentarism have returned to their values measured before the health crisis.

## Acknowledgments

We thank the French public health agency for access to the data.

## Author Contributions

**Formal analysis:** Jérémy Pierre.

**Methodology:** Jérémy Pierre, Cécile Collinet, Pierre-Olaf Schut, Charlotte Verdot.

**Software:** Jérémy Pierre, Charlotte Verdot.

**Writing – original draft:** Jérémy Pierre, Cécile Collinet, Pierre-Olaf Schut, Charlotte Verdot.

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
