## [Decision Letter · Decision Letter 0]

10 Mar 2022

PONE-D-21-09609A third of French older adults physically inactivePLOS ONE

Dear Dr. Pierre,

Thank you for submitting your manuscript to PLOS ONE. After careful consideration, we feel that it has merit but does not fully meet PLOS ONE’s publication criteria as it currently stands. Therefore, we invite you to submit a revised version of the manuscript that addresses the points raised during the review process.

Please, consider and answer in detail ALL comments from both reviewers.

We look forward to receiving your revised manuscript.

Kind regards,

Carlos Bueno Junior

Academic Editor

PLOS ONE

Journal Requirements:

3. Please modify the title to ensure that it is meeting PLOS’ guidelines (https://journals.plos.org/plosone/s/submission-guidelines#loc-title). In particular, the title should be "specific, descriptive, concise, and comprehensible to readers outside the field" and in this case we feel it is not informative and specific about your study's scope and methodology.

4. We note you have included a table to which you do not refer in the text of your manuscript. Please ensure that you refer to Table 6 in your text; if accepted, production will need this reference to link the reader to the Table.

Additional Editor Comments (if provided):

Please, consider and answer in detail ALL comments from both reviewers.

Reviewers' comments:

Reviewer's Responses to Questions

**Comments to the Author**

1. Is the manuscript technically sound, and do the data support the conclusions?

Reviewer #1: Yes

Reviewer #2: Yes

2. Has the statistical analysis been performed appropriately and rigorously? 

Reviewer #1: Yes

Reviewer #2: Yes

3. Have the authors made all data underlying the findings in their manuscript fully available?

Reviewer #1: Yes

Reviewer #2: Yes

4. Is the manuscript presented in an intelligible fashion and written in standard English?

Reviewer #1: Yes

Reviewer #2: No

5. Review Comments to the Author

Reviewer #1: Some linguistic corrections:

- in Table 5 and elsewhere, please use 'housework' or 'domestic work' rather than housekeeping

- unsure of use of 'point up' (denotes emphasis. 'point to' would suffice here

- Getting the public (no plural), or if you want to emphasise multiplicity, then use different population groups

- Avoid the use of 'the elderly' - Seniors or older adults is preferable

- use full stops for decimal points (not commas).

Requiring clarification:

- unsure what you mean by 'parent population' - total population?

- Reference to questionnaire administered face-to-face and self-administered. This isn't clear. Was the questionnaire delivered to people by someone who waited while respondents filled in it themselves. Or was the questionnaire in two parts, one part administered by a researcher and the other self-completed?

- what do you mean by 'complex sampling design'?

- I cannot find a special issue of Ageing in Society 2012. There is no journal with this title. However I think that Ageing & Society published a digital special issue on later life and physical activity. Is this what you mean?

Comments about the text:

- There is no reference to increasing recognition that the WHO PA recommendations are being discussed in the PA literature with a need identified to expand them with requirements to engage in strength/resistance, as well as balance and aerobic work. This reference might be useful 10.1136/bjsports-2018-100451

- I think the construction of a PA & SB typology should be signposted as an objective of the analysis. This typology enables you to integrate PA and SB into a more fine-grained understanding of older adults' practices and risks. This makes your analysis distinctive and should be promoted more forcefully in the discussion as well.

- Your discussion should return to gender differences - your findings appear to reflect the gendering of behaviours and dispositions as normative which culminate in greater or lesser propensity to PA & SB as men and women age. This is important.

- You claim in your conclusion that a third of older adults are sedentary - did you combine Type 1 and Type 2 (ie not PA & SB + not PA and not SB) to arrive at this claim? I wasn't sure. Please make explicit.

- You almost dismiss the finding that just over half are active and non-sedentary. That's quite an encouraging finding and perhaps should be highlighted as much as the disappointing % of non active, at risk older adults.

Reviewer #2: The study provides population-based estimates of self-reported physical activity and sedentary behaviour among older adults in France, and explores some of the potential sociodemographic and health correlates. The data are fairly recent (2014-2016), but things have likely become very different in recent years due to the Covid pandemic. This has been overlooked in the Discussion.

English language editing is recommended – especially consistency of tense.

Lack of page numbers make it difficult to locate comments/ suggestions

There are some referencing errors: eg in the paragraph “Screens: an activity popular among adults” there is a superscript reference as well as within-square-bracket references.

Abstract:

Study design unclear – longitudinal, cross-sectional?

Aim – very broad, without any actual aim/research question/hypothesis

Results – what does the P value refer to – women vs men? Results are not very informative.

Significant links with health indicators? Unclear which ones, in which way.

Conclusion – cause and effect cannot be determined by these data and analyses

Main paper:

Intro – PA definition is any movement, but then look at guidelines which are exclusively based on MVPA/VPA

Need reference for “these recommendations remain valid throughout life and are even more important with advancing age, which is generally associated with increasing vulnerability”

How were MET values for the physical activities from the survey determined?

Statistics: Presentation of study population descriptive statistics should be the first section of Results.

Results: Comparison with other studies/populations and discussion of potential mechanisms should occur in Discussion, or rename this section “Results and Discussion”

Change in PA over age – Authors say this is likely due to the deterioration in health, but there are no results presented to show deterioration in health with age.

Fig 1 Legend is in French language.

Throughout paper text and Tables: when presenting p values, unless p<0.001, please present exact values rather than “ns” or <0.01.

The wording “highest level of low PA” is a bit confusing. Suggest “lowest levels of PA”

Overall comment on Results: a lot of repetition of what is already presented in the Tables.

Conclusion – again, can’t say that you have measured effects on health. From the Results, there appears to only be one health indicator associated with sedentarism, which is obesity. I suggest the authors sharpen their aims so that they can present a clearer conclusion (and more concisely present Results) to address their aims/research questions.

6. PLOS authors have the option to publish the peer review history of their article (what does this mean?). If published, this will include your full peer review and any attached files.

Reviewer #1: No

Reviewer #2: No

---

## [Author Response · Author response to Decision Letter 0]

22 May 2022

Response to Reviewers:

We would like to begin by thanking the experts warmly for their valuable criticism. We have taken all their remarks into account in our revised version.

Responses to each comment can be found in the document entitled "Response to Reviewers".

The authors remain at your disposal for any questions you may have.

Yours sincerely

---

## [Decision Letter · Decision Letter 1]

27 Jul 2022

Physical activity and sedentarism among seniors in France, and their impact on health

PONE-D-21-09609R1

Dear Dr. Jérémy Pierre,

We’re pleased to inform you that your manuscript has been judged scientifically suitable for publication and will be formally accepted for publication once it meets all outstanding technical requirements.

Kind regards,

Carlos Bueno Junior

Academic Editor

PLOS ONE

Additional Editor Comments (optional):

Reviewers' comments:

Reviewer's Responses to Questions

**Comments to the Author**

1. If the authors have adequately addressed your comments raised in a previous round of review and you feel that this manuscript is now acceptable for publication, you may indicate that here to bypass the “Comments to the Author” section, enter your conflict of interest statement in the “Confidential to Editor” section, and submit your "Accept" recommendation.

Reviewer #1: All comments have been addressed

Reviewer #2: All comments have been addressed

2. Is the manuscript technically sound, and do the data support the conclusions?

Reviewer #1: Yes

Reviewer #2: Yes

3. Has the statistical analysis been performed appropriately and rigorously? 

Reviewer #1: Yes

Reviewer #2: Yes

4. Have the authors made all data underlying the findings in their manuscript fully available?

Reviewer #1: Yes

Reviewer #2: Yes

5. Is the manuscript presented in an intelligible fashion and written in standard English?

Reviewer #1: Yes

Reviewer #2: Yes

6. Review Comments to the Author

Reviewer #1: The authors have responded to my and Reviewer 2's comments and suggestions. I am satisfied that the paper is now of the appropriate standard for publication. Please correct mispelling of Bengsbo et al reference. Bengsbo, not Bangsbo.

Reviewer #2: (No Response)

7. PLOS authors have the option to publish the peer review history of their article (what does this mean?). If published, this will include your full peer review and any attached files.

Reviewer #1: No

Reviewer #2: **Yes: **Dorothea Dumuid

---

## [Editor Report · Acceptance letter]

1 Aug 2022

PONE-D-21-09609R1 

Physical activity and sedentarism among seniors in France, and their impact on health 

Dear Dr. Pierre:

I'm pleased to inform you that your manuscript has been deemed suitable for publication in PLOS ONE. Congratulations! Your manuscript is now with our production department. 

Kind regards, 

on behalf of

Dr. Carlos Bueno Junior 

Academic Editor

PLOS ONE